# CORTICAL-INSPIRED OPEN-BIGRAM REPRESENTATION FOR HANDWRITTEN WORD RECOGNITION

**Théodore Bluche**
A2iA SAS
Paris, France
tb@a2ia.com

**Christopher Kermorvant**
Teklia SAS
Paris, France
kermorvant@teklia.com

**Claude Touzet**
UMR CNRS NIA 7260
Aix Marseille Univ., Marseille, France
claude.touzet@univ-amu.fr

**Hervé Glotin**
UMR CNRS LSIS 7296
AMU, Univ. Toulon, ENSAM, IUF, France
glotin@univ-tln.fr

## ABSTRACT

Recent research in the cognitive process of reading hypothesized that we do not read words by sequentially recognizing letters, but rather by identifing open-bigrams, *i.e.* couple of letters that are not necessarily next to each other. In this paper, we evaluate an handwritten word recognition method based on original open-bigrams representation. We trained Long Short-Term Memory Recurrent Neural Networks (LSTM-RNNs) to predict open-bigrams rather than characters, and we show that such models are able to learn the long-range, complicated and intertwined dependencies in the input signal, necessary to the prediction. For decoding, we decomposed each word of a large vocabulary into the set of constituent bigrams, and apply a simple cosine similarity measure between this representation and the bagged RNN prediction to retrieve the vocabulary word. We compare this method to standard word recognition techniques based on sequential character recognition. Experiments are carried out on two public databases of handwritten words (Rimes and IAM), an the results with our bigram decoder are comparable to more conventional decoding methods based on sequences of letters.

## 1 INTRODUCTION

Taking inspiration in Biology is sometimes very efficient. For example, deep neural networks (NN) – which are outperforming all other methods (including support vector machines, SVM) in image recognition – are based on a series of several (usually 5 to 15) neurons layers, each layer involving sparsity in the activation pattern (a biological trait of the cortical map). The analogy continues with the modeling of the cortex as a hierarchy of cortical maps. Thanks to the analysis of reaction time in cognitive psychology experiments, the minimal number of cortical maps involved in a cognitive process is estimated to about ten, the same order of magnitude as the number of layers in deep neural networks for computer vision tasks. In the case of handwritten word recognition, Dehaene et al. have proposed a biologically plausible model of the cortical organization of reading (Dehaene et al., 2005) that assumes seven successive steps of increasing complexity, from the retinal ganglion cells to a cortical map of the orthographic word forms (Figure 1). One of the most recent successes of experimental psychology was the demonstration that human visual word recognition uses an explicit representation of letter position order based on letter pairs: the open-bigram coding (Whitney et al., 2012; Gomez et al., 2008; Grainger & Van Heuven, 2003; Glotin et al., 2010; Dufau, 2008).

As demonstrated in (Touzet et al., 2014), open-bigrams (OB) allow an over-coding of the orthographic form of words that facilitates recognition. OB coding favors same length words (i.e., neighbors of similar lengths). In the context of learning to read, the existence of the OB layer just before the orthographic word representation has been used to explain the lack of efficiency of whole language method (today banned from reading teaching) compared to the phonics method which explic-

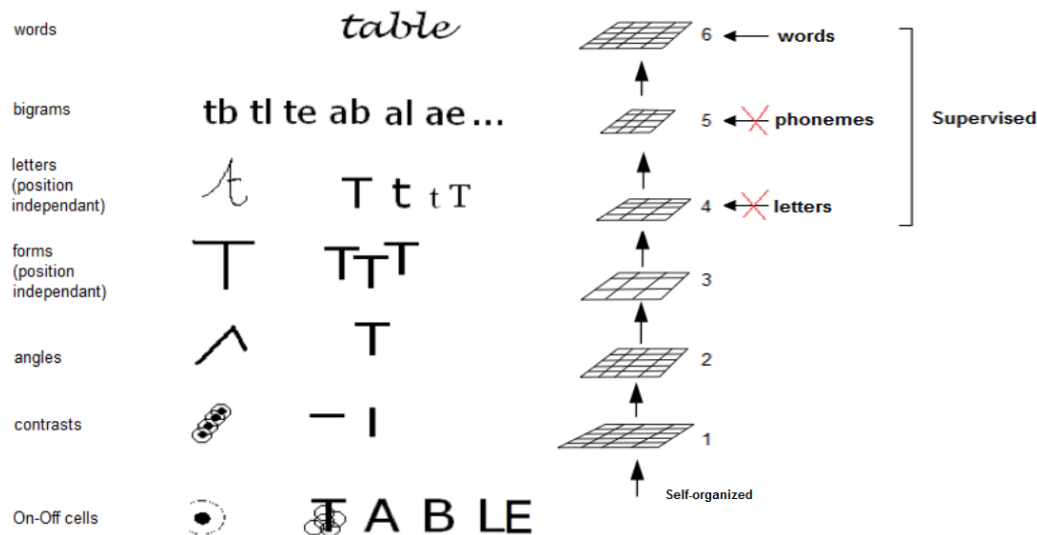

Figure 1: The cognitive process of reading, a seven steps procedure that includes an open-bigrams representation layer. Additional information helps the organization of levels 4 and 5, when using a phonics method, but not a whole language method (today banned from reading teaching for lack of efficiency, adapted from (Dehaene et al., 2005) and (Touzet, 2015).

itly supervises the organization of the OB map (with syllables), where the global method does not (Figure 1).

Since cognitive psychology has demonstrated the existence of the OB layer, the hypothesis has been put forward (Touzet et al., 2014) that the orthographic representation of words may have evolved in order to take into account the topology of the OB space, instead of the topology of the single letter space. Our goal here is to test this hypothesis, comparing OB vs sequential character recognition for word recognition. A state-of-the-art decoder based on a Long Short-Term Memory Recurrent Neural Networks (LSTM-RNN) is used on two public databases of handwritten words (Rimes and IAM).

The remaining of this paper will be divided as follows. In Section 2, we present related methods for handwritten word recognition. Then, we describe the open-bigram representation of words and the proposed decoder in Section 3. The experimental setup, including the data and the bigram prediction model, is explained in Section 4. Finally, we present our results in Section 5, before concluding in Section 6.

## 2 RELATED WORK

In this section, we give a brief overview of existing techniques for handwritten word recognition. Historically, the methods may be divided in three broad categories. The first approach is whole word recognition, where the image of the full word is directly classified into word classes, without relying on the character level (e.g. in (Parisse, 1996; Madhvanath & Govindaraju, 1996)). In the second method, the word image is segmented into parts of characters (stokes or graphemes). The segments are grouped and scored, and character sequences are obtained with a graph search (e.g. in (Bengio et al., 1995)) or with hidden Markov models (HMMs, e.g. in (Knerr et al., 1998)). The last method, most popular nowadays, is a segmentation-free approach. The goal is to predict a character sequence from the image without segmenting it first. The techniques include scanning a sliding window to extract features used in an HMM (e.g. in (Kaltenmeier et al., 1993)), or to feed the image to a neural network able to output sequences of character predictions (e.g. SDNNs (LeCun et al., 1998) or MDLSTM-RNN (Graves & Schmidhuber, 2009)).

More recently, different approaches have been proposed to recognize words using character bigrams, and therefore closer to the method we propose in this paper. Jaderberg et al. (2014) propose

to predict both the characters and ngrams of characters with two distinct convolutional neural networks (CNNs) to recognize text in natural images. Their approach includes a conditional random field as decoder. Similarly, Poznanski & Wolf (2016) train a CNN with a cross-entropy loss to detect common unigrams, bigrams or trigrams of character in a handwritten word image. The output of the network is matched against the lexicon using canonical correlation analysis. Almazán et al. (2014) use Fisher vectors from images and pyramidal character histograms, to learn a feature space shared by the word images and labels, for word spotting, also using canonical correlation analysis.

## 3 PROPOSED METHOD

### 3.1 AN OPEN-BIGRAM REPRESENTATION OF WORDS

The letter bigrams of a word $w$ is the set of pairs of consecutive letters. The **open-bigram of order** $d$ is the set of pairs of letters separated by $d$ other letters in the word, which we call $\mathcal{B}_d(w)$:

$$\mathcal{B}_d(w) = \{w_i w_{i+d} : i \in \{1 \ldots |w| - d\}\}. \tag{1}$$

The usual bigrams are open-bigrams of order 1. By extension, we call $\mathcal{B}_0(w)$ the set of letters in the word $w$. For example, for word word, we have:

$$\mathcal{B}_1(\texttt{word}) = \{\texttt{or}, \texttt{rd}, \texttt{wo}\} \ ; \ \mathcal{B}_2(\texttt{word}) = \{\texttt{od}, \texttt{wr}\} \ ; \ \mathcal{B}_3(\texttt{word}) = \{\texttt{wd}\}.$$

The general open-bigram representation of a word is the union of

$$\mathcal{B}_{d_1,\ldots,d_n}(w) = \mathcal{B}_{d_1}(w) \cup \ldots \cup \mathcal{B}_{d_n}(w). \tag{2}$$

For example, $\mathcal{B}_{1,2,3}(\texttt{word}) = \{\texttt{od}, \texttt{or}, \texttt{rd}, \texttt{wd}, \texttt{wo}, \texttt{wr}\}$.

We extend $\mathcal{B}$ into $\mathcal{B}'$ by including special bigrams for the letters at the beginning and end of a word:

$$\mathcal{B}'(w) = \mathcal{B}(w) \cup \{\_w_0, w_{|w|}\_\}. \tag{3}$$

So, for example,

$$\mathcal{B}'_{1,2,3}(\texttt{word}) = \{\_\texttt{w}, \texttt{d}\_, \texttt{od}, \texttt{or}, \texttt{rd}, \texttt{wd}, \texttt{wo}, \texttt{wr}\}. \tag{4}$$

In this paper, we will call $B$ the set of all bigrams, and $W$ the set of all words. We will represent a word of the vocabulary $w \in W$ as a normalized binary vector $\mathbf{v}_{w \in W} \in \Re^{|B|}$

$$\mathbf{v}_w = \frac{[\delta(b \in \mathcal{B}(w))]_{b \in B}}{\sqrt{|\mathcal{B}(w)|}}, \tag{5}$$

*i.e.* the vector with $0$ everywhere and $1/\sqrt{|\mathcal{B}(w)|}$ at indices corresponding to bigrams of the word. The stacking of the vector representation of all the words in the vocabulary yields the vocabulary matrix $V \in \Re^{|W| \times |B|}$.

Note that in this representation, the bigrams form an unordered set. We *do not know*: (i) where the bigrams are, (ii) what is the order of a given bigram, (iii) how many times it occurs. The goal is to build a word recognition decoder in the bigram space.

### 3.2 AN OPEN-BIGRAM DECODER

While the trivial representation of a word is an ordered sequence of letters, the order in the bigram space is locally embedded in the bigram representation. Most state-of-the-art word recognition systems recognize sequences of letters, and organize the vocabulary for a constrained search as directed graphs, such as prefix trees, or Finite-State Transducers. On the other hand, we can interpret the bigram representation as encoding directed edges in a graph, although we will not explicitly build such a graph for decoding.

On Figure 2, we show the graph for a representation of the word into a sequence of letters. Gray edges show the potential risk of a misrecognition in the letter sequences. On Figure 2(b), we display the conceptual representation of bigrams as edges. We observe that a global order of letters can emerge from the local representation. Moreover, the constituent information of a word in the

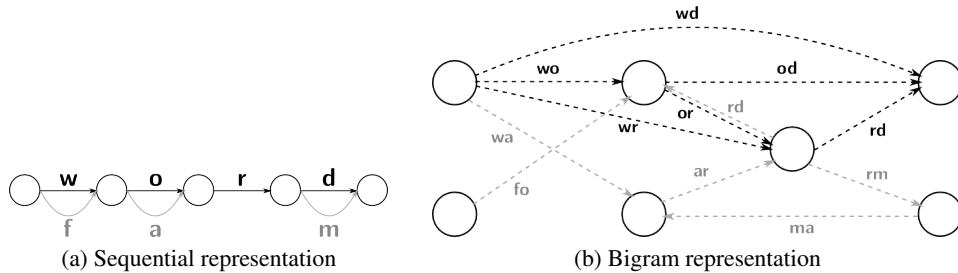

(a) Sequential representation (b) Bigram representation

Figure 2: Word representation as an explicit sequence of letters (a), and as a set of bigrams (b). Grey edges show the potential impact of misrecognitions.

bigram space is redundant, potentially making this representation more robust to mispredictions of the optical model.

The optical model is the system which provides the predictions of bigrams from the image (or, in the classical approach sequences of character predictions). That is, it provides a confidence measure that each bigram $b$ is present in image $x$: $0 \leq p_b(x) \leq 1$. This is transformed into a vector in the bigram space:

$$\mathbf{q}_x = \frac{[p_b(x)]_{b \in B}}{\sqrt{\sum_b p_b^2(x)}}. \tag{6}$$

For decoding, we chose the very simple cosine similarity between the query ($\mathbf{q}_x$) and a vocabulary word ($\mathbf{v}_w$). Since we normalized both vectors, this is simply the dot product:

$$d(\mathbf{q}_x, \mathbf{v}_w) = \mathbf{v}_w^T \mathbf{q}_x, \tag{7}$$

so the similarity with all words of the vocabulary can be computed with a matrix-vector product:

$$D_V(x) = V^T \mathbf{q}_x. \tag{8}$$

The recognized word is the one with maximum similarity with the query:

$$w^* = arg \max D_V(x) = arg \max_w \frac{\sum_{b \in \mathcal{B}(w)} p_b(x)}{\sqrt{|\mathcal{B}(w)| \sum_b p_b^2(x)}}. \tag{9}$$

We carried out a few preliminary experiments to justify the open-bigram decoder. First, we considered the famous sentence with mixed up letters:

> *"aoccdrnig to a rscheearch at cmabrigde uinervtisy it deos not mttaer in waht oredr the ltteers in a wrod are the olny iprmoatnt tihng is taht the frist and lsat ltteers be at the rghit pclae the rset can be a toatl mses and you can sitll raed it wouthit porbelm tihs is bcuseae the huamn mnid deos not raed ervey lteter by istlef but the wrod as a wlohe".*

Although the origin and validity of this statement when letters are put in the right order has been discussed [1], it is true that most of us can read it without trouble. For each word of more than one letter in this sentence, we computed the open-bigram representation ($d = 0..3$), and replaced it with the word having the highest cosine similarity in the English vocabulary described in the next section. The result was:

> *"according to a **researcher** at **abridged** university it does not matter in what **ordered** the letters in a word are the only important thing is that the first and last letters be at the right place the rest can be a total **messes** and you can still read it **outwith** problem this is because the human mind does not read every letter by itself but the word as a whole".*

Note that the word "cambridge" was not in the vocabulary. Although the task in this paper is not to recognize mixed up words, it shows the ability of our decoder to perform a reading task that we naturally do.

---

[1] http://www.mrc-cbu.cam.ac.uk/people/matt.davis/cmabridge/

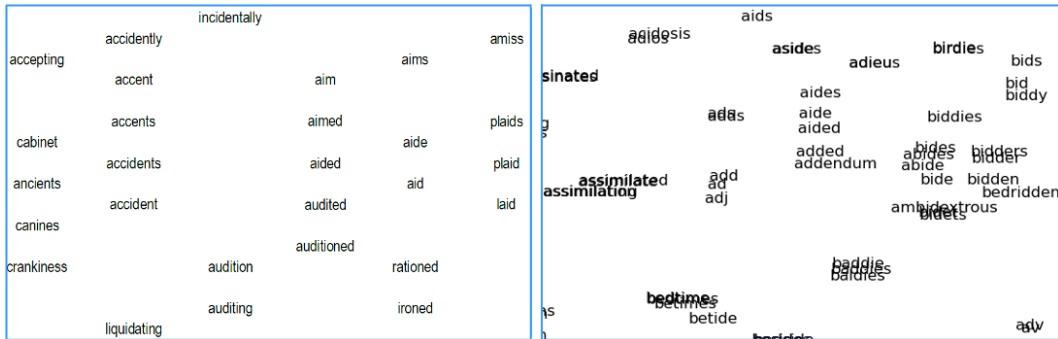

Figure 3: Visualization of the bigram representation of the English vocabulary, for $d = 1..3$ (Touzet et al., 2014) (left), *vs* after t-SNE (Van der Maaten & Hinton, 2008) (right). Our complete bigramic map of English: `https://youtu.be/OR2vjj8MNeM?t=197`.

On Figure 3, we show the English vocabulary in bigram space ($d = 1..3$), reduced to two dimensions with t-SNE (Van der Maaten & Hinton, 2008). We observe that words which are close in the bigram space also have a close orthographic form.

# 4 EXPERIMENTAL SETUP

## 4.1 DATA PREPARATION

We carried out the experiments on two public handwritten word databases: Rimes (Augustin et al., 2006) (French), and IAM (Marti & Bunke, 2002) (English). We simplified the problem by limiting ourselves to words of at least two lowercase characters (*a* to *z*). This selection removed approximately 30% of the words. The number of words and bigrams of different orders in the different sets are reported on Table 4, in Appendix A.1.

We applied deslanting (Buse et al., 1997), contrast enhancement, and padded the images with 10px of white pixels to account for empty context on the left and right of words. From the preprocessed images, we extracted sequences of feature vectors with a sliding window of width 3px. The features are geometrical and statistical features described in (Bianne et al., 2011), which give state-of-the-art results in handwritten text line recognition (Bluche et al., 2014).

We downloaded word frequency lists for French and English[2]. These lists were built from film subtitles [3] written by many contributors, and they contain many misspellings. We removed the misspelled words using GNU Aspell (Atkinson).

We selected 50,000 words for each language. They are the most frequent words (length $\geq 2$) and made only of lowercase characters between *a* and *z*, making sure to also include all the words of the database. For example, the 50,000 most frequent French words fulfilling these condition miss about 200 words of the Rimes database, so we selected the most frequent 49,800 and added the missing 200. Note that most of the words that were removed from the dataset are not shorter words, but words with special or upper case characters. The distribution of lengths of filtered-out words is shown on Figure 5 in Appendix A.1.

## 4.2 RECOGNITION OF OPEN-BIGRAMS WITH RECURRENT NEURAL NETWORKS (RNNS)

To predict bigrams, we chose Bidirectional Long Short-Term Memory RNNs (BLSTM-RNNs) for their ability to consider the whole sequence of input vectors to make predictions. We trained one RNN for each order-$d$ bigram, with the Connectionist Temporal Classification (CTC (Graves et al., 2006)) criterion. The CTC framework defines a sequence labeling problem, with an output sequence of labels, of smaller length than the input sequence of observations.

---

[2]`http://invokeit.wordpress.com/frequency-word-lists/`.
[3]`http://opensubtitles.org`

We built the target sequences for training as sequences of bigrams, ordered according to the first letter of the bigram. For example, for $d = 2$, the target for `example` is `ea-xm-ap-ml-pe`. The CTC training criterion optimizes the Negative Log-Likelihood (NLL) of the correct label sequence. We set the learning rate to 0.001, and stopped the training when the NLL on the validation set did not decrease for 20 epochs. We kept the network yielding the best NLL on the validation set.

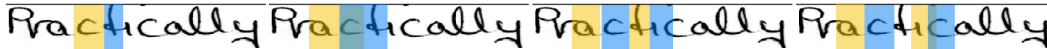

Figure 4: Hypothetical context needed in the input image to make two consecutive (yellow and blue) bigram predictions, for $d = 0$ (left, to predict `c`, then `t`) to 3 (right, to predict `ai`, then `cc`). As $d$ increases, the contexts become more complex to model: they involve long range dependencies and are highly intertwined.

We trained one RNN for each order $d = 0$ to 3, including the special bigrams for word extremities or not. We will refer to each of these RNNs with $rnn_d$ for order $d$ ($rnn_{d'}$ when extremities are included). The architecture of the networks is described in Appendix A.3. These RNNs are trained to predict sequences of fixed order bigrams. Here, we are interested in a word representation as a bag of bigrams, which does not carry any information about the sequence in which the bigrams appear, the number of times each bigram appears, or the order of each individual bigram. That is, we are interested in a decoder which considers an unordered set of bigrams predictions across bigram orders.

We **forget the temporal aspect** of bigram predictions by taking the maximum value of a given bigram prediction by the RNN (where $rnn_d(x, t)$ if the output of the RNN for order $d$, input image $x$ at timestep $t$):

$$p_{d,b}(x) = \max_t rnn_d(x, t), \tag{10}$$

and we **forget the bigram order** by taking the maximum output across different values of $d$:

$$p_b(x) = \max_d \max_t rnn_d(x, t). \tag{11}$$

It would have been more satisfying for this experiment to train an optical model to predict a set of bigrams for all orders. However, this work is focused on the decoder. Moreover, even the simpler task of predicting a sequence of bigrams of fixed order is challenging (the sequence error rates of these networks are detailed in Appendix B.2). On Figure 4, we show the hypothetical context needed to make two consecutive predictions, for bigram order $d = 0..3$. RNNs are popular for handwriting recognition, and can consider a context size of variable length – but still local – to predict characters ($d = 0$).

For $d = 1$, the required context is still local (and would span two consecutive characters), but overlap, because each character is involved in two bigrams. For $d > 1$, the context is even split into two areas (covering the involved characters) that might be far apart depending on $d$. Contexts for different predictions are entangled: the whole area between two characters forming a bigram is not relevant for this bigram (and might be of varying size), but will be important to predict other bigrams. It means that the RNN will have to remember a character observation for some time, until it sees the second character of the bigram, while ignoring the area in between for this bigram prediction, but remembering it since it will be useful in order to predict other bigrams. The number of classes for bigrams is also 26 times larger than the number of characters, making the classification problem harder, and the number of examples per class in training smaller.

## 5 RESULTS

In this paper, we focused on a subset of Rimes and IAM word databases, which makes the comparison with published results difficult. Instead, we compared the bigram decoder approach to decoding with standard models, consisting of a beam search with Viterbi algorithm in the lexicon. However, these standard models yield state-of-the art results on the reference task for the two considered database (Bluche et al., 2014).

## 5.1 BASELINE SYSTEMS BASED ON HMMS AND VITERBI DECODING

We built several models and used the same vocabulary as for the bigram decoder, and no language model (all words have the same prior probability). These baseline systems are based on HMMs, with emission models made either of Gaussian mixtures (GMM/HMM), Multi-Layer Perceptrons (MLP/HMM) or Recurrent Neural Networks ($rnn_0$/HMM). They are almost identical to those presented in a previous work (Bluche et al., 2014), where a comparison is made with state-of-the-art systems for handwritten text line recognition. More details about these models and their training procedure are presented in Appendix A.2.

Table 1: Word Error Rates (%) with baseline systems and Viterbi decoding of character sequences.

| | | | Models | |
|---|---|---|---|---|
| | Dataset | GMM/HMM | MLP/HMM | $rnn_0$/HMM |
| **Rimes** | Valid. | 37.38 | 14.82 | 10.79 |
| *Viterbi (Char. seq.)* | Test | 36.24 | 14.45 | 10.03 |
| **IAM** | Valid. | 27.64 | 11.73 | 10.21 |
| *Viterbi (Char. seq.)* | Test | 37.96 | 19.97 | 17.49 |

On Table 1, we report the percentages of word errors on the validation and test sets of Rimes and IAM. The best word error rates are around 10% (17.5% on the test set of IAM), and constitute the baseline performance to which the bigram approach is to be compared.

## 5.2 MEASURING THE QUALITY OF BIGRAM PREDICTIONS

Since we keep a confidence value for all bigrams in the prediction vector, rather than using a binary vector (cf. Eq. 6), we modified the formulation of precision and recall. A bigram $b \in \mathcal{B}(w)$ is correctly retrieved with confidence $p_b(x)$, and missed with confidence $(1 - p_b(x))$. Similarly, a bigram not in the representation $\mathcal{B}(w)$ of word $w$ is falsely recognized with confidence $p_b(x)$, and correctly ignored with confidence $(1 - p_b(x))$. It gives us the following expressions for precision and recall

$$precision = \frac{\sum_{(x,w)} \sum_{b \in \mathcal{B}(w)} p_b(x)}{\sum_x \sum_{b' \in B} p_b(x)}, \qquad recall = \frac{\sum_{(x,w)} \sum_{b \in \mathcal{B}(w)} p_b(x)}{\sum_{w \in W} |\mathcal{B}(w)|}, \qquad (12)$$

which are the usual ones when $p_b(x) \in \{0, 1\}$. The *F-measure* is calculated from precision and recall with the usual formula.

Table 2: Precision, Recall and F-measure of OB detection by RNNs with different orders $d$.

| | | 0 | 1 | 1' | 2 | 2' | 3 | 3' | 1,2,3 | 1',2',3' |
|---|---|---|---|---|---|---|---|---|---|---|
| | | | | | | $d$ | | | | |
| **Rimes** | Precision | 95.0 | 89.9 | 91.2 | 79.8 | 82.8 | 74.8 | 82.6 | 84.5 | 84.0 |
| | Recall | 93.4 | 87.6 | 89.3 | 84.8 | 85.8 | 83.4 | 80.9 | 86.7 | 88.5 |
| | F-measure | 0.94 | 0.89 | 0.90 | 0.82 | 0.84 | 0.79 | 0.82 | 0.89 | 0.86 |
| **IAM** | Precision | 93.5 | 87.3 | 89.3 | 77.7 | 81.6 | 62.3 | 76.2 | 80.5 | 81.0 |
| | Recall | 92.5 | 86.2 | 88.5 | 82.3 | 84.0 | 77.5 | 78.6 | 84.3 | 86.4 |
| | F-measure | 0.93 | 0.87 | 0.89 | 0.80 | 0.83 | 0.69 | 0.77 | 0.82 | 0.84 |

The results for all RNNs, and for the combination of orders, are reported on Table 2. We observe that the precision and recall results are correlated to the performance in terms of edit distance or sequence error rates. Namely, they decrease as the bigram order increases, which is not surprising, given that higher order bigrams are more difficult to recognize with these sequence models. We also see that including the special bigrams for word beginnings and endings generally improves the results. This is not surprising either: the RNNs are good at recognizing them.

Despite this performance decrease, the precision remains above 70%, which limits the amount of noise that will be included in the bigram representation for recognition. Combining the recognition across orders, we obtain a precision of around 84% on Rimes and 80% on IAM. The recall tends to be higher than the precision, staying around or above 80% in all configurations. Across orders, the

recall is above 88% on Rimes and 86% on IAM. The high recall will limit the amount of missing information in the bigram representation.

Overall, the F-measure for bigram recognition is above 80%, which is a good starting point, given that *(i)* the vocabulary used in decoding will add constraints and may help recovering from some mistakes in the bigram recognition, and *(ii)* the redundancy and order encoded in the bigram may limit the impact of misrecognitions.

## 5.3 WORD RECOGNITION USING BIGRAM PREDICTIONS

On Table 3, we report the results of bigram decoding. For each word image in the validation and test sets, we computed the bigram predictions with the RNNs described above. We combined the different orders as explained previously, and either added the special bigrams for word boundaries and/or the single character predictions or not. We computed the cosine similarity to the bigram decomposition of all words in the vocabularies in the same representation space (*i.e.* same orders, and same choices for the inclusion of special bigrams and single characters) by computing the product of the vocabulary matrix $V$ by the recognition vector. We counted the number of times the correct word was not the most similar one.

Table 3: Decoding results (% of word errors).

| Decoding | Model | Rimes | | IAM | |
|---|---|---|---|---|---|
| | | Valid | Test | Valid | Test |
| Viterbi *(Char. seq.)* | Best in Table 1 | 10.79 | 10.03 | 10.21 | 17.49 |
| *Cosine (bigrams)* | $rnn_{1,2,3}$ | 25.58 | 24.37 | 13.45 | 20.82 |
| | $rnn_{1',2',3'}$ | 12.43 | 12.27 | 11.80 | 19.25 |
| | $rnn_{0,1,2,3}$ | 11.03 | 10.41 | 11.98 | 19.61 |
| | $rnn_{0,1',2',3'}$ | 9.81 | 9.43 | 11.09 | 18.39 |

We see that adding the special bigrams for word boundaries improves the results, especially when single characters are not included in the representation. A possible explanation, besides the fact that they tend to be recognized more easily, could be that they provide a very useful information to disambiguate words having a similar bigram representation (*e.g.* `them` and `theme`). Adding single characters also improves the performance of the decoder, especially when the boundary bigrams are not included in the representation. The gain obtained with the single characters is about the same – sometimes a little better – as the gain with boundaries. It might be due to the much better recognition of the RNN for single characters (precision and recall over 90%), as well as the added redundancy and complementary information provided. The results of decoding with different combinations of orders are presented in the appendices in Table 7. They confirm those observations. The best performance is achieved with both single characters and word boundaries, although the gain compared to adding only one of them is slight. The error rates are competitive or better than the best error rates obtained by classical character sequence modeling and Viterbi decoding.

## 6 CONCLUSION

State-of-the-art systems, as well as most of the systems for handwritten word recognition found in the literature, either try to model words as a whole, or as a sequence of characters. The latter, which currently gives the best results, is widely adopted by the community, and benefits from a lot of attention. In this paper, we have proposed a simple alternative model, inspired by the recent findings in cognitive neurosciences research on reading.

We focused on the representation of words in the open-bigram space and built an handwritten word recognition system operating in that space. We were interested in observing how a simple decoding scheme, based on a mere cosine similarity measure in the bigram space, compared to traditional methods. The main apparent difficulty arises from the fact that the global ordering of characters and the distance between bigram constituents are lost in this representation.

The qualitative results presented in the first section showed that the envisioned approach was viable. With the letter reordering example, we have seen that the correct orthographic form of words can be retrieved with a limited and local knowledge of character orders. Moreover, we validated that words

that are close in orthographic form are also close in the bigram space. Thus, we demonstrated that the open-bigram representation shows interesting and competitive metric properties for the word recognition. Current work consists in learning most discriminant open-bigram at different order, possibly higher than three according to the length of the word and its similarity to others.

ACKNOWLEDGMENTS

This work was conducted in COGNILEGO project 2012-15, supported by the French Research Agency under the contract ANR 2010-CORD-013 http://cognilego.univ-tln.fr.

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

# A DATA AND MODELS

## A.1 DATA

Table 4: Number of words/bigrams in different sets of Rimes and IAM databases. (in parentheses, the number of distinct tokens).

|  |  | Train | Valid. | Test |
|---|---|---|---|---|
| **Rimes** | Words | 33,947 | 3,772 | 5,695 |
|  | Bigram $d = 0$ | 161,203 (26) | 17,887 (25) | 26,828 (25) |
|  | Bigram $d = 1$ | 127,256 (312) | 14,115 (240) | 21,133 (258) |
|  | Bigram $d = 2$ | 93,309 (425) | 10,343 (336) | 15,438 (352) |
|  | Bigram $d = 3$ | 68,747 (436) | 7,651 (327) | 11,372 (368) |
|  | Bigram $d = 1..3$ | 289,312 (517) | 32,109 (420) | 47,943 (452) |
| **IAM** | Words | 38,584 | 5,977 | 18,394 |
|  | Bigram $d = 0$ | 177,101 (26) | 26,133 (26) | 82,320 (26) |
|  | Bigram $d = 1$ | 138,517 (428) | 20,156 (364) | 63,926 (417) |
|  | Bigram $d = 2$ | 99,933 (550) | 14,179 (477) | 45,532 (537) |
|  | Bigram $d = 3$ | 68,961 (543) | 9,361 (457) | 30,537 (522) |
|  | Bigram $d = 1..3$ | 307,411 (598) | 43,696 (551) | 139,995 (592) |

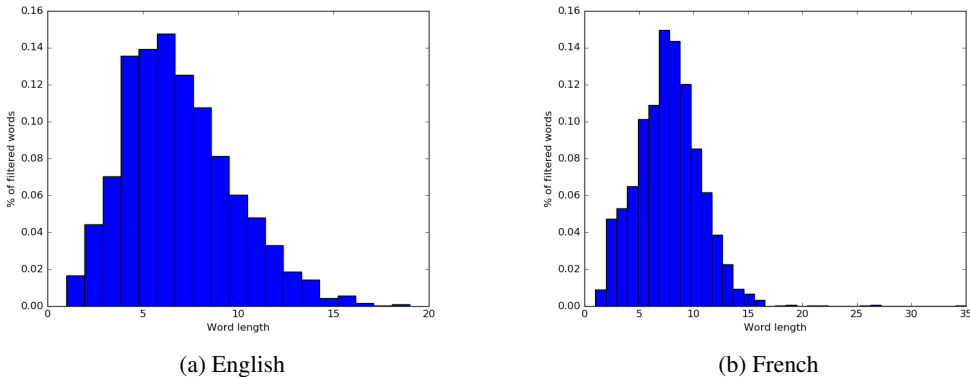

(a) English (b) French

Figure 5: Distribution of lengths of filtered-out words

## A.2 BASELINE SYSTEMS

We built several models and used the same vocabulary as for the bigram decoder, and no language model (all words have the same prior probability). The first one is a Hidden Markov Model, with 5 (Rimes) or 6 (IAM) states per characters, and an emission model based on Gaussiam Mixture Models. This system is trained with the Maximum Likelihood criterion, following the usual Expectation-Maximization procedure. At each iteration, the number of Gaussians is increased, until no improvement is observed on the validation set. The forced alignments with the GMM/HMM system are used to build a labeled dataset for training a Multi-Layer Perceptron (MLP) with 4 hidden layers of 1,024 sigmoid units. We optimized the cross-entropy criterion to train the network to predict the HMM states from the concatenation of 11 input frames, with a learning rate of 0.008. The learning rate was halved when the relative improvement was smaller than 1% from one epoch to the next. The MLP is integrated in the hybrid NN/HMM scheme by dividing the predicted state posteriors $p(s|x)$ by the state priors $p(s)$, estimated from the forced alignments. It is then further trained with a sequence-discriminative criterion: state-level Minimum Bayes Risk (Kingsbury, 2009) (sMBR) for 5 epochs. Finally, we also performed the sequential decoding with vocabulary constraints using $rnn_0$. The input features for all systems are the same, described in 4.1. The results of similar

systems for handwritten text line recognition can for example be found in a previous work (Bluche et al., 2014), where a comparison is made with state-of-the-art systems.

## A.3 RECURRENT NEURAL NETWORK

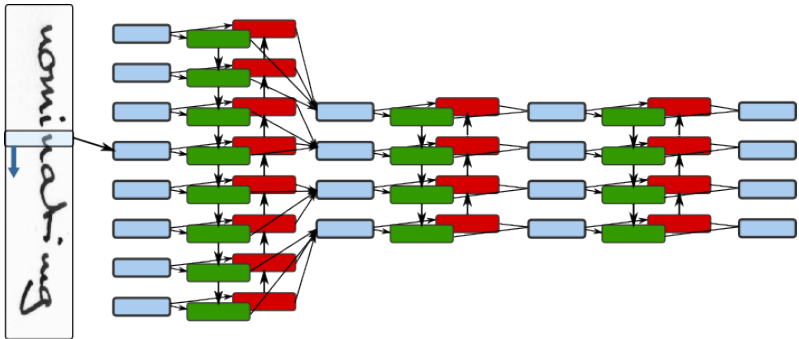

Figure 6: Neural network architecture. A sequence of feature vectors is extracted with a sliding window, and fed to a multi-layer BLSTM network, with subsampling after the first BLSTM layer. The output is a sequence of open-bigram prediction.

The RNNs, depicted on Figure 6, have seven hidden layers, alternating Long Short-Term Memory (Hochreiter & Schmidhuber, 1997) recurrent layers in both direction and feed-forward layers. The first LSTM layers have 100 hidden LSTM units. The BLSTM outputs of two consecutive timesteps of both directions are fed to a feed-forward layer with 100 nodes, halving the size of the sequence. The next LSTM layers have 150 units, and are connected to a feed-forward layer with 150 units. The last two LSTM layers and feed-forward layer have 200 hidden units.

The inputs of the networks are sequences of geometrical and statistical features (described in (Bianne et al., 2011)), extracted with a sliding window of width 3px. The outputs are sequences of open-bigram predictions.

The networks are trained with stochastic gradient descent to minimize the Connectionist Temporal Classification (CTC (Graves et al., 2006)) criterion, i.e. the Negative Log-Likelihood (NLL) of the correct label sequence. We set the learning rate to 0.001, and stopped the training when the NLL on the validation set did not decrease for 20 epochs, and kept the network yielding the best NLL on the validation set.

Table 5: Number of parameters in the RNNs

| $d$ | Num. Parameters |
|---|---|
| 0 | 1,805,827 |
| 1 | 2,066,477 |
| 1' | 2,087,329 |
| 2 | 2,066,477 |
| 2' | 2,087,329 |
| 3 | 2,066,477 |
| 3' | 2,087,329 |
| 1,2,3 | 6,199,431 |
| 1',2',3' | 6,261,987 |

## B  RESULTS

### B.1  CORRELATION BETWEEN CHARACTER EDIT DISTANCES AND OB COSINE SIMILARITIES

On Figure 7, we randomly selected pairs of words, and pairs of words with high cosine similarity in the French and English dictionary, and plotted their cosine similarity in the bigram space against

the edit distance between the two words, normalized by the length of the longest word. We note that words with high cosine similarity also have short edit distance, supporting the idea that the bigram representation encode some global letter order, and therefore might favor word recognition.

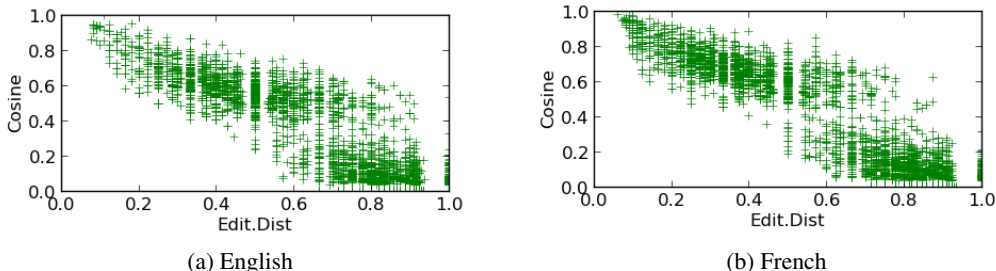

(a) English                           (b) French

Figure 7: Relation between cosine similarity in bigram space and normalized edit distance, for the English (a) and French (b) vocabularies.

## B.2 EVALUATION OF PERFORMANCE OF RNN ALONE

We trained RNNs to predict sequences of bigrams of a given order. Their performance to accomplish this task can be measured, on the validation set, with the edit distance (edit.dist) between the recognized sequence and the true sequence, and with the percentage of sequences with at least one error: the Sequence Error Rate (SER).

Table 6: RNN results

|       | $d$ | edit.dist(%) | SER(%) | Precision(%) | Recall(%) | F-measure |
|-------|-----|--------------|--------|--------------|-----------|-----------|
| **Rimes** | 0 | 8.3 | 22.7 | 95.00 | 93.42 | 0.942021 |
|       | 1 | 11.4 | 20.8 | 89.86 | 87.61 | 0.8872 |
|       | 1' | 9.6 | 21.7 | 91.17 | 89.25 | 0.901998 |
|       | 2 | 13.4 | 23.2 | 79.78 | 84.84 | 0.822315 |
|       | 2' | 11.1 | 26.2 | 82.84 | 85.84 | 0.843128 |
|       | 3 | 14.8 | 23.7 | 74.80 | 83.37 | 0.788523 |
|       | 3' | 12.5 | 27.4 | 82.57 | 80.93 | 0.817402 |
|       | 1,2,3 | - | - | 84.53 | 86.68 | 0.855922 |
|       | 1',2',3' | - | - | 84.03 | 88.48 | 0.86197 |
| **IAM** | 0 | 8.0 | 21.7 | 93.51 | 92.54 | 0.930205 |
|       | 1 | 13.1 | 23.2 | 87.34 | 86.20 | 0.867681 |
|       | 1' | 10.4 | 23.3 | 89.28 | 88.48 | 0.888813 |
|       | 2 | 16.7 | 26.2 | 77.71 | 82.33 | 0.799537 |
|       | 2' | 12.7 | 28.3 | 81.57 | 83.95 | 0.827408 |
|       | 3 | 20.8 | 31.2 | 62.29 | 77.54 | 0.69081 |
|       | 3' | 14.6 | 31.8 | 76.17 | 78.56 | 0.773436 |
|       | 1,2,3 | - | - | 80.53 | 84.26 | 0.823527 |
|       | 1',2',3' | - | - | 81.04 | 86.40 | 0.836315 |

The results are reported on Table 6. We observe that the performance decreases as $d$ increase. Yet the errors remain in a reasonable range compared to the simple case $d = 0$, and one should keep in mind that a small degradation should be expected from the larger number of classes and smaller number of training examples per class.

We have shown that RNNs can perform the apparently difficult task of recognizing sequences of open-bigrams. Thus, we may use the RNN bigram predictions in our decoder, rather than building bigram predictions from letter ones, which would have been less satisfying considering the supposed interest of the redundancy of the bigram representation that allows error corrections.

## B.3 Evaluation of Performance of the decoder

In Table 7, we report the results of the cosine decoder applied to the outputs of OB RNNs with different bigram orders.

Table 7: Decoding result (word error rate) with different bigram orders

| $d$ | Rimes | IAM |
|---:|:---:|:---:|
| 1 | 14.5 | 16.4 |
| 0,1 | 13.0 | 14.1 |
| 1' | 12.9 | 13.1 |
| 0,1' | 11.9 | 12.3 |
| 1,2 | 26.4 | 13.7 |
| 1',2' | 12.7 | 11.8 |
| 0,1,2 | 11.7 | 12.3 |
| 0,1',2' | 10.1 | 11.3 |
| 1,2,3 | 25.6 | 13.5 |
| 1',2',3' | 12.4 | 11.8 |
| 0,1,2,3 | 11.0 | 12.0 |
| 0,1',2',3' | 9.8 | 11.1 |

In Table 8, we evaluate the cosine decoder by applying it to the ground-truth OB decomposition with varying order of the validation sets, i.e. the performance assuming the RNN optical models are perfect.

Table 8: Decoding result (word error rate) with different bigram orders

| $d$ | Rimes | IAM |
|---:|:---:|:---:|
| 1 | 0.64 | 1.86 |
| 0,1 | 0.90 | 1.62 |
| 1' | 0.03 | 0.02 |
| 0,1' | 0.03 | 0.02 |
| 1,2 | 0.21 | 0.05 |
| 1',2' | 0.03 | 0.00 |
| 0,1,2 | 0.13 | 0.05 |
| 0,1',2' | 0.29 | 0.00 |
| 1,2,3 | 0.21 | 0.08 |
| 1',2',3' | 0.03 | 0.00 |
| 0,1,2,3 | 0.03 | 0.03 |
| 0,1',2',3' | 0.03 | 0.00 |

## B.4 Error Analysis

### B.4.1 Comparison of Errors of the Sequential and Open-Bigram Models

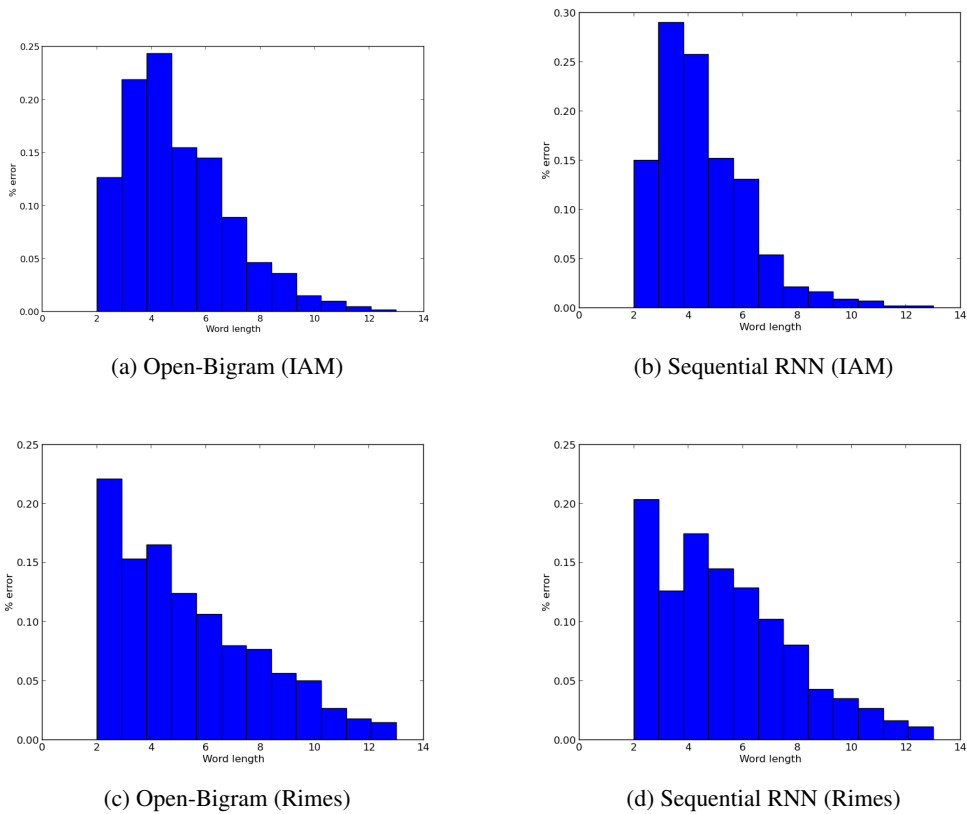

(a) Open-Bigram (IAM)

(b) Sequential RNN (IAM)

(c) Open-Bigram (Rimes)

(d) Sequential RNN (Rimes)

Figure 8: Distribution of word errors vs. word length for the proposed (left) and sequential (right) approaches. We observe the same behaviour: short words tend to be more difficult to recognize.

