# Peer review of "Cortical-Inspired Open-Bigram Representation for Handwritten Word Recognition"

_ICLR 2017 — rejected_

[Official Review · AnonReviewer2 · rating 7 · confidence 5 · 16 Dec 2016]

This submission investigates the usability of cortical-inspired distant bigram representations for handwritten word recognition. Instead of generating neural network based posterior features for character (optionally in local context), sets posterior for character bigrams of different length are used to represent words.  The aim here is to investigate the viability of this approach and to compare to the standard approach.

Overall, the submission is well written, although information is missing w.r.t. to the comparison between the proposed approach and the standard approach, see below.

It would be desirable to see the model complexity of all the different models used here, i.e. the number of parameters used.

Language models are not used here. Since the different models utilize different levels of context, language models can be expected to have a different effect on the different approaches. Therefore I suggest to include the use of language models into the evaluation.

For your comparative experiments you use only 70% of the data by choosing longer words only. On the other hand, it is well known that the shorter words are more prone to result in misrecognitions. The question remains, if this choice is advantageous for one of the tasks, or not - corresponding quantitative results should be provided to be able to better evaluate the effect of using this constrained corpus. Without clarification of this I would not readily agree that the error rates are competitive or better than the standard approach, as stated at the end of Sec. 5.

I do see the motivation for introducing open-bigrams in an unordered way due to the corresponding evidence from cognitive research. However, decision theoretically I wonder, why the order should be given up, if the underlying sequential classification problem clearly is of a monotonous nature. It would be interesting to see an experiment, where only the use of the order is varied, to differentiate the effect of the order from the effect of other aspects of the approach.

End of page 1: "whole language method" - please explain what is meant by this.

Page 6: define your notation for rnn_d(x,t).

The number of target for the RNNs modeling order 0 (unigrams effectively) and the RNNs modeling order 1 and larger are very much different.  Therefore the precision and recall numbers in Table 2 do not seem to be readily comparable between order 0 and orders >=1. At least, the column for order 0 should be visually separated to highlight this.


Minor comments: a spell check is recommended
p. 2: state-of-art -> state-of-the-art
p. 2: predict character sequence -> predict a character sequence
p. 3, top: Their approach include -> Their approach includes
p. 3, top: an handwritten -> a handwritten
p. 3, bottom: consituent -> constituent
p. 4, top: in classical approach -> in the classical approach
p. 4, top: transformed in a vector -> transformed into a vector
p. 5: were build -> were built
References: first authors name written wrongly: Thodore Bluche -> Theodore Bluche

[Official Review · AnonReviewer3 · rating 4 · confidence 4 · 19 Dec 2016]

This paper uses an LSTM model to predict what it calls "open bigrams" (bigrams of characters that may or may not have letters inbetween) from handwriting data. These open bigrams are subsequently used to predict the written word in a decoding step. The experiments indicate that the system does slightly better than a baseline model that uses Viterbi decoding. I have some major concerns about this paper:

- I find the "cortical inspired" claim troublesome. If anything, it is psychology/cognitive science inspired, in the sense that open bigrams appear to help for word recognition (Touzet et al. 2014). But the implied cortical characteristics, implicitly referred to e.g. by pointing to analogies between deep neural nets for object recognition and in that case the visual cortex, is unfounded. Is there any direct evidence from neuroscience that open-bigrams constitute a wholly separate layer in the cortex for a handwriting recognition task? Dehaene's work is a proposal, so you'll need to describe more "findings in cognitive neurosciences [sic] research on reading" (p. 8) to substantiate those claims. I am further worried by the fact that the authors seem to think that "deep neural networks are based on a series of about five pairs of neurons [sic] layers". Unless I misunderstand something, you are specifically referring to Krizhevsky's AlexNet here (which you should probably have cited there)? I hope you don't mean to imply that all deep neural nets need five layers. It is also not true that ten is "quite close to the number of layers of an efficient deep NN" -- what network? what task? etc.

- The model is not clearly explained. There is a short paragraph in Appendix A.3. that roughly describes the setup, but this does not include e.g. the objective function, or answer why the network output is only considered each two consecutive time steps, rather than at each time step (or so it seems?). This is probably because the paper argues that it "is focused on the decoder" (p. 6), rather than on the whole problem. I find this problematic, because in that case we're effectively measuring how easy it is to reconstruct a word from its open bigrams, which has very little to do with handwriting recognition (it could have been evaluated on any text corpus). In fact, as the example on page 4 shows, handwriting is not necessary to illustrate the open bigram hypothesis. Which leads me to wonder why these particular tasks were chosen, if we are only interested in the decoding mechanism?

- The comparison is not really fair. The Viterbi decoder only has access to unigrams, as far as I can tell. The only model that does better than that baseline has access to a lot more information, and does not do that much better. Did the Viterbi model have access to the word boundary information (at one point rather confusingly called "extremities") that pushed the open bigram model over the edge in terms of performance? Why is there no comparison to e.g. rnn_0,1' (unigram+bigram+boundary markers)? The dataset also appears to be biased in favor of the proposed approach (longer words, only ). I am not convinced that this paper really shows that open bigrams help.

I very much like the idea of the paper, but I am simply not convinced by its claims.

Minor points:
- There are quite a few typos. Just a sample: "independant" (Fig.1), "we evaluate an handwritten", ", hand written words [..], an the results", "their approach include", "the letter bigrams of a word w is", "for the two considered database"
- Wouldn't it be easy to add how many times a bigram occurs, which would improve the decoding process? You can just normalize over the full counts instead of the binary occurrence counts.
- The results in Table 5 are the same (but different precision) as the results in Table 2, except that edit distance and SER are added, this is confusing.

[Official Review · AnonReviewer1 · rating 5 · confidence 5 · 20 Dec 2016 (modified: 23 Jan 2017)]

This paper explores the use of Open Bigrams as a target representation of words, for application to handwriting image recognition. 

Pros:
- The use of OBs is novel and interesting.
- Clearly written and explained.

Cons:
- No comparison to previous state of the art, only with author-generated results. 
- More ablation studies needed -- i.e. fill in Table3 with rnn0,1 rnn0,1,2 rnn0,1' etc etc. It is not clear where the performance is coming from, as it seems that it is single character modelling (0) and word endings (') that are actually beneficial.
- While the use of Open bigrams is novel, there are works which use bag of bigrams and ngrams as models which are not really compared to or explored. E.g.

[Final Decision · Program Chairs · 06 Feb 2017]
**ICLR committee final decision**

There is consistent agreement towards the originality of this work and that the topic here is "interesting". Additionally there is consensus that the work is "clearly written", and (excepting questions of the word "cortical") all would be primed to accept this style of work. 
 
 However there is a shared concern about the quality and potential impact of the work, in particularly in terms of the validity of empirical evaluations. Reviewers are generally not inclined to believe that the current empirical evidence validates the conclusions of the word. Suggestions are to: make greater use of a language model, compare to external baselines, or remove the handwriting aspects.